# Development and In Vitro and In Vivo Evaluation of an Antineoplastic Copper(II) Compound (Casiopeina III-ia) Loaded in Nonionic Vesicles Using Quality by Design

**DOI:** 10.3390/ijms232112756

**Published:** 2022-10-22

**Authors:** Zenayda Aguilar-Jiménez, Mauricio González-Ballesteros, Silvia G. Dávila-Manzanilla, Adrián Espinoza-Guillén, Lena Ruiz-Azuara

**Affiliations:** Departamento de Química Inorgánica y Nuclear, Facultad de Química, Universidad Nacional Autónoma de México, Av. Universidad 3000, Mexico City DF 04510, Mexico

**Keywords:** nanocarriers, quality by design, niosome, cancer, Casiopeina III-ia, copper

## Abstract

In recent decades, the interest in metallodrugs as therapeutic agents has increased. Casiopeinas are copper-based compounds that have been evaluated in several tumor cell lines. Currently, casiopeina III-ia (CasIII-ia) is being evaluated in phase I clinical trials. The aim of the present work is to develop a niosome formulation containing CasIII-ia for intravenous administration through a quality-by-design (QbD) approach. Risk analysis was performed to identify the factors that may have an impact on CasIII-ia encapsulation. The developed nanoformulation optimized from the experimental design was characterized by spectroscopy, thermal analysis, and electronic microscopy. In vitro drug release showed a burst effect followed by a diffusion-dependent process. The niosomes showed physical stability for at least three months at 37 °C and 75% relative humidity. The in vitro test showed activity of the encapsulated CasIII-ia on a metastatic breast cancer cell line and the in vivo test of nanoencapsulated CasIII-ia maintained the activity of the free compound, but showed a diminished toxicity. Therefore, the optimal conditions obtained by QbD may improve the scaling-up process.

## 1. Introduction

The use of metallodrugs for the treatment of cancer has been of great interest in recent years. Even though platinum-based compounds are widely used for chemotherapy, there is still a low interest in the clinical use of metal-based drugs. This is attributed to their inherent toxicity. In this respect, the use of nanomedicine can be an alternative to reduce their toxic effects. Peña et al., reported that from the total number of articles published involving “nano”/“chemotherapy”/“metals”, 40% corresponded to essential metals, including copper which accounted for 10% of the search [1]. For example, cisplatin is one of the most used compounds for the treatment of several types of cancer. Nevertheless, the adverse effects have led to a search for diverse alternatives, such as endogenous metal-based drugs that include the copper-based compounds, casiopeinas [2]. On the other hand, casiopeinas are a family of more than 100 copper mixed-chelate coordination compounds with polypharmacological activities [3,4,5,6]. These compounds have been evaluated in cultures of murine tumor cell lines of melanoma, glioma, and leukemia. Additionally, they have shown activity in several human tumor cell cultures (colon, cervical, breast, lung, neuroblastoma, Lewis lung, and glioblastoma multiforme carcinoma) and presented efficacy in in vivo tests of xenograft tumor models with a colon carcinoma HCT-15 tumor [7,8,9]. CasIII-ia [Cu(4,4’-dimethyl-2,2´-bipyridine)(acetylacetonate)(NO_3_)] stands out amongst this group due to its solubility in water and lower toxicity than the more active analogs in preclinical studies [6]. Moreover, it exhibited a wide therapeutic range; thus, it was selected to start in a clinical phase I trial in Mexico [10,11]. Pharmacokinetic studies made in Wistar rats and rabbits presented low drug blood concentration. Additionally, protein binding studies in human plasma proved that a decrease in CasIII-ia concentrations leads to an increase in protein binding [12,13]. The compiled background allowed us to propose the nanoencapsulation of CasIII-ia. In this sense, when CasIII-ia was loaded into chitosan nanoparticles [14], it was observed that CBF1/Hsd mice transplanted with B16 melanoma tumors had an increased life span compared to the animals which received the free drug. However, a high polydispersity particle size was obtained (150 nm–10 µm). Additionally, the encapsulation percentage was low due to the hydrophilicity of CasIII-ia [14].

Niosomes, like liposomes, are vesicular systems that can encapsulate both hydrophilic and lipophilic drugs. Niosomes are formed mainly by a nonionic surfactant that is less expensive than phospholipids and chemically stable; unlike phospholipids, surfactants do not require specific handling conditions [15]. One of the most widely used surfactants is Span 60^®^ (S) since it allows a greater drug encapsulation efficiency [16,17]. Besides the surfactant, cholesterol (C) may increase the rigidity of the membrane, whereas Pluronic F127 (P) improves the physical stability of the vesicles [18,19].

Furthermore, the nanoformulation of metallodrugs represents one of the main challenges in the development of nanoparticles due to the metal’s capacity to react with the nanocarriers, which modifies the overall properties of the drug [1]. Therefore, the physicochemical and structural characteristics of the drug to be encapsulated must be considered. In addition, understanding the conditions and general factors involved in the final design of nanoparticles will allow modulating their physicochemical and biologic properties [20,21]. To have consistency in those properties, the design, synthesis, manufacturing, and formulation of the pharmaceutical product must follow a systematic approach that uses quality-by-design (QbD) tools [22,23,24]. In the last five years, studies on the use of tools involved in quality by design (QbD) have been reported for platinum and vanadium [25,26].

Therefore, the present work proposes a nanoformulation consisting of niosomes bearing CasIII-ia designed and optimized with QbD tools for its IV administration. The advantages of the application of QbD in niosome formation and metal-based drug encapsulation is underlined.

## 2. Results

### 2.1. Quality-by-Design Approach

#### 2.1.1. Risk Analysis (RA)

Among the quality target product profile (QTPP) elements, critical quality attributes (CQAs) and critical process parameters (CPPs), as well as the critical material attributes (CMAs) in the initial RA, were performed. The Ishikawa diagram (Figure 1) allowed the identification of all the potential factors of formulation and process parameters that influence the CQAs. The factors were collected and grouped into three sections (based on CPPs and CMAs, from the Ishikawa diagram), as they are related to the niosome formulation (drug and material properties), niosome preparation (ether injection method), and uncontrollable variables (relative humidity, room temperature, and residual solvent). After the identification of the risk with the Ishikawa diagram, a risk analysis was performed with RRMA, which is a qualitative tool for linking the likelihood of occurrence with the severity of harm to identify the impact of CMAs and CPPs on CQAs. Table 1 and Table 2 present the estimated interrelationships of the selected critical factors for CMA and CPP factors. High risk represents risk that is not acceptable due to safety/efficacy issues; medium risk is acceptable but needs investigation; and low risk means no further investigation is needed [27]. The classification of the impact of each factor was realized based on previous reports, which allowed us to obtain knowledge of space development and identify the principal high-risk factors that may impact the formulation of metal-based drug encapsulation in niosomes. Then, variables were evaluated in experimental designs to find which had the greatest impact on the CQAs. The variables were divided into two categories: formulation and method of production. The variables obtained were as follows: CasIII-ia concentration; S/C/P concentration; Pluronic F127 concentration; time during evaporation; speed during evaporation; temperature injection; injection speed; speed during injection; and solvent volume.

#### 2.1.2. Experimental Design

Once the risk analysis was performed and the main variables were defined, experimental designs were carried out to evaluate the effects of each factor on responses. The Plackett–Burman design was used to screen variables, whereas the factorial design and central composite design helped to predict and optimize the nanoformulation.

##### Plackett–Burman Design

Variables defined from the Plackett–Burman design (Appendix A) were analyzed through Pareto charts of standardized effects, normalized tests, residue graphs, and ANOVAs. Variables are organized according to the effect: the highest effects appear beyond the red lines in the Pareto diagrams for particle size, PDI, surface charge, and encapsulation efficiency. Three main variables were selected according to their statistical significance (*p* < 0.05) and more frequent effect on the responses (observed in Pareto charts). Based on the results, drug concentration, niosome concentration, and the speed rate during injection were selected to be evaluated in a second design.

##### Factorial Design 2^k^

The optimization of the nanoformulation of a factorial design 2^3^ was executed. The three main variables obtained by the Plackett–Burman design were carried out in a two-level factorial design with three factors (Appendix A). Pareto charts and variance analysis allowed us to identify which variables and interactions are statistically significant (Appendix A).

Variables and their interactions, on the CQAs, can be expressed quantitatively through a regression model. Then, linear regression equations were obtained for each response evaluated to obtain the best values of niosome formation from the desired values of CQAs. In general, for all cases (except for PDI), a good fit to the linear model was obtained where R^2^ for each CQA was 97.35%, 75.15%, 97.77%, and 92.91% for particle size, PDI, EE, and surface charge, respectively. From the factorial design, central points can be added to evaluate whether a quadratic model might have a better fit for the data obtained from the experimental runs through a central composite design (CCD).

##### Central Composite Design

The CCD is obtained from an extension of the factorial design to which axial points are added to make the adjustment of the quadratic model possible with the linear variables, two-variable interaction, and quadratic terms of the continuous variables. The CCD helps to better understand and optimize the response. Eight experimental runs (two were central points and six were axial points) were added to the factorial design previously made. The twenty runs previously made (factorial design) and the new eight runs (CCD) performed were analyzed through Minitab 17 (Appendix A). In the surface plot (Figure 2), the interactions between the variables are observed and those that are statistically significant were obtained by the analysis of variance, where it was observed that all factors influence particle size; however, for the other responses it was found that only the CasIII-ia concentration had a *p*-value < 0.05.

Quadratic equations helped to describe the individual main effect and the interaction effect of factors on the responses (CQAs). From the analysis of variance, residual, and normality plots (Appendix A) it was determined that the quadratic equations obtained are adequate for the prediction and optimization of the nanoformulation (Appendix A). Based on the results, an optimization plot (Appendix A), obtained from the Minitab software, was used for identifying the combination of variable settings that jointly optimize the required values of CQAs (particle size of 150 nm and maximum encapsulation efficiency).

### 2.2. Preparation of Optimized Nanoformulation

The validation of the applied QbD methodology was confirmed by selecting the operational conditions to prepare the optimized formulation of niosomes with CasIII-ia (Table 3).

Table 4 shows the results (predicted and observed values) obtained from the optimized conditions. It is possible to observe that the optimal conditions of particle size (150 nm) and encapsulation efficiency (40%) were obtained. Results show that experimental values are similar to predicted values.

### 2.3. Optimized Nanoformulation Characterization

#### 2.3.1. Electronic Microscopy

Spheric niosome morphology was observed with electronic microscopy. CasIII-ia inside the aqueous part of the vesicle is observed through the images obtained by STEM (without negative staining) (Figure 3a). In both TEM and STEM micrographs, the presence of the compound inside (TEM) and in the periphery (STEM) of the vesicles is indicated within the lighter areas due to the contrast generated by the copper atoms. Energy-dispersive spectroscopy (EDS) confirmed the presence of copper (Appendix A). In the micrographs obtained by TEM (with negative staining) (Figure 3b), bright regions (light) can be observed in the vesicles that could indicate that the uranyl acetate is immersed in the center of the vesicle (acoustical region), demonstrating that the vesiculation was carried out. Moreover, particle sizes around 150 nm were observed, which is consistent with the results observed by dynamic light scattering (Appendix A).

#### 2.3.2. Thermal Analysis

From the optimized nanoformulation, DCS and TGA (Figure 4) were carried out for the niosome without CasIII-ia, niosome with CasIII-ia, and free CasIII-ia. In DSC, a small endothermic peak is observed for CasIII-ia around 85 °C due to the dehydration of CasIII-ia, and this was confirmed by TGA with the weight loss of the two water molecules. The DSC curve of niosomes without CasIII-ia shows a broad endothermic peak at 42 °C, which is due to conformational changes and the subsequent loss of crystallinity in the Span 60, cholesterol, and Pluronic F127 molecules during the vesiculation [42,43]. Incorporation of CasIII-ia in niosomes produced a displacement of the transition peak at 47 °C. The foregoing may be associated with the existence of CasIII-ia within the niosome, which could change the conformation of vesicles. The endothermic peak in niosomes at 105 °C is associated with the incorporation of Pluronic in the niosomal system. When the niosome does not contain Pluronic F127, this peak is not observed. Niosomes with CasIII-ia showed the highest intensity peak at 105 °C, which may be due to the interaction of CasIII-ia with the niosome membrane. This supports the hypothesis that CasIII-ia molecules are found interacting with the niosomal membrane. An exothermic peak at 241 °C was observed in DSC for free CasIII-ia, corresponding to the degradation of the acetylacetonate ligand of CasIII-ia as reported elsewhere [14]. The latter was confirmed by TGA through weight loss. When the niosome with CasIII-ia was evaluated, the peak at 241 °C was not observed in either DSC or TGA. This behavior was also observed when the CasIII-ia was encapsulated in chitosan and was attributed to the incorporation of CasIII-ia inside the vesicle [14]. Moreover, the organic degradation (of niosome compounds) was observed at 380 °C in TGA.

#### 2.3.3. FTIR Studies

FTIR spectra (Figure 5) showed in the red line the characteristic bands of bipyridine and acetylacetonate of CasIII-ia (at 1616 cm^−1^, vibrations correspond to the C = N bond of bipyridine; at 1525 cm^−1^, vibrations C = C; at 1587 cm^−1^, corresponding to the carbonyl group of the acetylacetonate). On the other hand, in the niosome spectra, stretching vibrations of the O-H bond (Span 60 and Pluronic F127) are observed at 3430 cm^−1^. Bands at 2920–2850 cm^−1^ are attributed to the C-H stretch vibrations of the saturated aliphatic group (Span 60 and Pluronic) and in the band at 1731 cm^−1^, the carbonyl stretches of the aliphatic ester functional group correspond to Span60. The band at 1384 cm^−1^ corresponds to the nitrate ion; this band appears in both spectra (CasIII-ia and niosome with CasIII-ia), whereas the characteristic bands of CasIII-ia are not observed in the niosome spectra with CasIII-ia, except nitrate ion, which is probably due to the encapsulation of the compound in concordance with the results obtained in the thermograms. However, through FTIR-ATR (Appendix A), it was observed that the characteristic bands of C = N in diimines in the CasIII-ia are present after the encapsulation of the compound, indicating a possible presence of the compound in the niosome membrane, which could explain the decrease in the zeta potential of the niosome when CasIII-ia is incorporated. Chitosan with CasIII-ia in FTIR studies does not show the characteristic bands of the drug inside the chitosan nanoparticle. This suggests that CasIII-ia can be loaded into chitosan nanoparticles [14].

#### 2.3.4. Stability Studies

Accelerated stability studies (40 °C and 75% RH with *n* = 3, according to NOM-073-SSA1-2015) were carried out for niosomes with and without CasIII-ia. The physical stability of the samples was evaluated by particle size, zeta potential, and polydispersity index (Figure 6). Following the results, after three months, for niosomes and niosomes with CasIII-ia, there was a decrease in particle size and zeta potential, and a decrease in the polydispersity index only for niosomes with CasIII-ia. The decrease in particle size was statistically significant only for Nio/CasIII-ia.

#### 2.3.5. In Vitro Drug Release

The release profile of CasIII-ia (Figure 7) through the nanocarrier was biphasic with a fast release (burst effect) in the first hours, followed by a slower release for up to 33 h in accordance with a prolongated release. To evaluate the release profile of CasIII-ia from the niosome, the data obtained were fitted to zero-order, first-order, Higuchi model, and Korsmeyer–Peppas model kinetics (Appendix A). The results indicated that the best fit was obtained with the Korsmeyer–Peppas model, meaning that the release of CasIII-ia is mainly governed by a Fick diffusion mechanism. The latter has been reported for drugs that are encapsulated in liposomes and niosomes [44,45].

#### 2.3.6. Cytotoxicity Assay

In vitro assays were performed on a triple-negative breast cancer tumor line (MDA-MB-231). Encapsulated CasIII-ia and the niosome blank were evaluated at different concentrations at 24 or 48 h after treatment, using the sulforhodamine B method [46]. Figure 8 shows a slight decrease in the activity of CasIII-ia as a product of encapsulation, compared to free CasIII-ia in both timeframes. However, encapsulated CasIII-ia reached 50% of inhibition at a lower concentration when exposed to the cells for 48 h than for 24 h. In contrast, the niosome without CasIII-ia showed no activity at the concentrations and times evaluated.

#### 2.3.7. In Vivo Assays

In vivo study assays of CasIII-ia were performed in female BALB/c mice (*n* = 7). Cis-Pt was used as a control. Nonencapsulated CasIII-ia, nanoencapsulated CasIII-ia, and cis-Pt decreased the tumor growth speed (Figure 9a). Additionally, a decrease in the final weight of the primary tumor (Figure 9b) was observed, with all three treatments being statistically significant with respect to the untreated group.

In addition, the weight of the mice was evaluated at the end of the treatment, where it was observed that only the groups treated with the niosome with CasIII-ia and niosome without CasIII-ia recovered their post-treatment weight. The effect was statistically significant with respect to the group treated with CasIII-ia. (Figure 10a), indicating lower toxicity. This was evaluated using the percentage of weight loss, as an indicator of toxicity (Figure 10b), which was calculated as follows: [[weight at day 21/weight at day 0] − 1] × 100. An unpaired *t*-test found that only the niosome-treated group was statistically significant concerning the untreated group. Despite the niosome with CasIII-ia being not statistically significant, this group shows a trend of lower toxicity.

Figure 10a shows there is less weight loss with the encapsulated CasIII-ia regarding the nonencapsulated CasIII-ia and cis-Pt. There was no significant difference between the niosome target and the 5% glucose solution, which could indicate low niosome activity.

## 3. Discussion

Nanocarriers can increase the rate to cross the gap for metallodrugs in clinical applications. The selection of carriers that are simple, easily scalable, approved, and compatible with the target metallodrugs will facilitate the clinical translation [1]. For the water-soluble compound CasIII-ia, a niosomal system with already approved and unreactive excipients (Span 60, cholesterol, and Pluronic F127) was used. The presence of the hydrophilic head and hydrophobic chain of Span 60 results in the formation of a double-layer structure that provides the host for hydrophilic and hydrophobic drugs. Niosomes are nanocarriers with a wide range of applications, an are mainly used in nanomedicine [47]. Niosomes contain a high number of variables for their formulation; therefore, strict control over variables of nanocarriers is required for optimizing the metallodrug formulation. In this sense, the use of quality-by-design tools allows for the optimization of metallodrug formulation through the definition of CQAs that impact QTPPs. For example, particle size and encapsulation efficiency (CQAs) are important factors for IV administration (QTPPs). In this sense, particle sizes between 100 and 200 nm are required, as they have been reported to show adequate fenestration through blood vessels [48,49]. High encapsulation efficiencies could decrease the binding of drugs to proteins. Consequently, the variables that impact the principal CQAs were evaluated in a screening design (Plackett–Burman). Subsequently, factorial and central composite designs were utilized for formulation optimization. The experimental designs indicate that drug concentration is the key factor in the design of these niosomes. We observed that the particle size and encapsulation efficiency, principally, depend on the molecular weight of the compound to be nanoencapsulated, as reported by previous works [50]. S/C/P concentration and speed rate during injection have a lower impact on particle size and encapsulation efficiency. The behavior of the CQAs was modeled as a function of the three variables. A formulation with the operational conditions for a particle size of 150 nm and maximum encapsulation efficiency was realized. Experimental values are in accord with predicted values, indicating that the formulation is repeatable and reproducible. In addition, equations obtained may predict other formulations within the design space. For particle size, the predicted and experimental value was 150 nm. However, as was observed, the particle size increased when the niosomes were without CasIII-ia (215 nm); such an effect is the opposite of what has been reported for nanoencapsulated hydrophilic compounds. Usually, it is observed that when hydrophilic drugs are encapsulated, the particle size increases [51]. The decrease in particle size in the presence of CasIII-ia may be due to the presence of hydrogen bond interactions between the copper compound and the hydrophilic heads of Span 60 and Pluronic F127 chains; however, this hypothesis remains to be confirmed. Additionally, CasIII-ia molecules could be forming intermolecular interactions, leading to a decrease in particle size. CasIII-ia loaded into chitosan nanoparticles presents the same behavior, but has an encapsulation efficiency of 6% [14]. In this work, the efficiency encapsulation obtained of 40% may be due to the steric hindrance that could generate the chains of Pluronic F127 where the molecules are probably competing with the chains of the polymer and forming weak interactions with them. The latter would explain the obtained zeta potential of −13.8 mV compared to the zeta potential of −23.4 mV of the niosomes without CasIII-ia. Thus, we hypothesize that CasIII-ia may be on the surface of the vesicles, which diminishes the zeta potential. Computational methods such as DFT or particle dissipative dynamics would help to confirm the hypotheses of the interactions between drug molecules and those that make up the niosome; at the moment, this part is beyond the scope of this work. The intermolecular interaction and drug encapsulation had an influence on the modified drug delivery and the burst effect. Considering that 40% encapsulation was obtained, the burst effect could be due to the nonencapsulated CasIII-ia rapidly passing through the dialysis bag. Subsequently, a slower release of nonencapsulated CasIII-ia was found to form weak interactions with the niosome membrane. The longest release was a consequence of the encapsulated CasIII-ia. In contrast, almost 90% of the free-control CasIII-ia diffused out of the dialysis bag in the first hour of the study. The latter is characteristic of vesicular systems, such as liposomes and niosomes [52,53].

The increase in the in vitro activity of encapsulated CasIII-ia with respect to time can be correlated with the drug release tests where a modified release was observed. It indicates that the longer the time, the greater the complete delivery of the drug and, therefore, the greater the activity of the compound. Shaker and collaborators mention that the cytotoxic effect in MCF-7 cells (breast cancer) of tamoxifen loaded in niosomes is affected by the release time of the compound from the niosomal vesicles [33]. The nonactivity in the MDA-MB-231 tumor line of niosomes shown in in vitro assays is consistent with what was observed in in vivo studies. These results agree with Agarwal et at., who showed that niosomes have low activity in cancer cell lines and normal cells [54]. On the other hand, Haroun and collaborators observed lower toxicity (in breast cancer) in niosomes functionalized with PEG loaded with brucine (natural alkaloid) in comparison with free brucine [31]. They attributed this behavior to the enhanced and permeation retention effect, which increases the possibility of the niosomes reaching the site of action. In addition, the functionalization with PEG provides a longer circulation time of the niosomes [31]. The same behavior is observed with nanoparticles containing Pluronic F127 in the formulation.

In regard to the antitumor activity, cis-Pt and encapsulated and free-control CasIII-ia showed similar activities; however, cis-Pt showed the highest toxicity considering the frequency of administration: every 7 days, four doses (cis-Pt) vs. every 4 days, six doses (encapsulated and free-control CasIII-ia). 

The optimized and predicted formulation was obtained and characterized. The spheric shape of niosomes observed in TEM and STEM is characteristic of the vesicular system [52]. Results showed that the thickness of the membrane formed by the Span 60/cholesterol/Pluronic F127 is around 50 nm. This bilayer formation has also been observed by microscopy in other works [55]. 

Physical stability studies showed a decrease in zeta potential and in particle size. Copolymer Pluronic F127 is temperature-dependent. A rise in temperature at 40 °C could cause the dehydration of hydrophobic polypropylene oxide (PPO) blocks, causing the polymer chains to potentially come closer. The latter would cause the particle size and zeta potential to decrease in the blank niosome and niosome with CasIII-ia. A further decrease in the zeta potential in niosomes with CasIII-ia could be attributable to reaching an equilibrium between CasIII-ia and the niosome membrane. It has been observed that a negative zeta potential changes the value for niosomes with Pluronic L64 based on the temperature [56]. Although the particle size decreased and was statistically significant, the size is found to be between 100 and 150 nm, which is suitable for its use in intravenous administration [57]. Despite the decrease in the zeta potential, the decrease in particle size indicates there is no coalescence of the nanoparticles due to the steric hindrance caused by the long poly(propylene oxide) and poly(ethylene oxide) chains of Pluronic F127, which indicates physical stability for the niosomes with CasIII-ia for at least three months. These results agree with the reports made by other authors where niosomes are stable for at least 90 days [58,59]. 

## 4. Materials and Methods

### 4.1. Materials

Casiopeina^®^ III-ia was synthesized at the National Autonomous University of Mexico. CAS [223930–33–4], the copper (II) complex was prepared following the reported patent [60]. Span 60^®^ (S), cholesterol (C), and Pluronic F127^®^ (P) were purchased from Sigma-Aldrich (Ciudad de México, Mexico). The analytical grade ethylic ether was purchased from Baker (Ciudad de México, Mexico). All reagents were used without further purification. 

### 4.2. Methods

#### 4.2.1. Quality-by-Design Approach: Risk Analysis

The Ishikawa diagram was constructed to identify all potential variables (formulation and formation method), which can have an impact on the CQAs (particle size, encapsulation efficiency, zeta potential, and polydispersity index) of the product, and therefore, alter the QTPPs. Later, a relative risk-based matrix analysis (RRMA) was performed based on a qualitative tool to identify the impact of an individual variable obtained from the Ishikawa diagram. The interdependence of the attribute of each CPP and CMA between CQAs was classified into high-, medium-, and low-risk attributes based on the impact on the finished product quality [27].

##### Quality Target Product Profile (QTPP)

QTPPs define the relationship between quality, security, and efficacy. Outlining the QTPP will define the CQAs, affecting the final product quality considerably. Table 5 describes the QTPP of nanoencapsulated CasIII-ia.

##### Critical Quality Attributes (CQAs)

The CQAs defined consider the IV administration, the physical and chemical characteristics of CasIII-ia, and the security of the product. CQAs are presented in Table 6.

##### Critical Process Parameters (CPPs) and Critical Material Attributes (CMAs)

CPPs and CMAs are variable parameters of the process that affect the CQAs and should be monitored or controlled for quality. CPPs and CMAs (from raw materials and niosome formation) that could have a high impact on the CQA (particle size, EE, PDI, and superficial charge) were identified in the Ishikawa diagram.

#### 4.2.2. Niosome Formation

Once the profile, parameters, and attributes were defined, the niosomes were prepared using a modified ether injection method [28,69] described as follows: Span 60 and cholesterol (1:1 molar, [70,71]) were dissolved in ethyl ether. Pluronic F127 and CasIII-ia were dissolved in distilled water. The ether dissolution was added into a volume of CasIII-ia/Pluronic F127 solution water drop by drop using a syringe pump (kdScientific Model 100 series) on a magnetic stirrer. Then, the organic solvent was removed by a slow magnetic stirrer at room temperature for one hour. The suspension was filtered through a syringe filter (Merck MillexTM, PES 0.22 µm). A blank of niosome was prepared in the same way. The conditions for Plackett–Burman and factorial design niosome formation are presented in Appendix A. For total Span 60/Cholesterol/Pluronic F17 (S/C/P) concentrations: Pluronic F127 was used in a range of 2.5–5% molar; cholesterol and Span 60 in a range of 47.5–48.5% molar each.

#### 4.2.3. Quality-by-Design Approach: Experimental Design (DoE)

Once the procedure of the niosome formation was recognized, Plackett–Burman and factorial designs were used to screen the variables and optimize the nanoformulation. Experimental designs were created and analyzed with Minitab 17 statistical software (State College, PA, USA). For all experimental designs, the niosomes were prepared according to Section 4.2.2.

##### Plackett–Burman Design

Nine variables were selected through the Ishikawa diagram. The variables were screened through a Plackett–Burman design to identify the most important factors that affect the CQAs. The factors evaluated were: time of evaporation (A), speed rate during evaporation (B), temperature of injection (C), injection rate (D), speed rate during injection (E), solvent volume (F), CasIII-ia concentration (G), S/C/P concentration (H), and P concentration (I). The responses evaluated were particle size (Y_1_), polydispersity index (Y_2_), superficial charge (Y_3_), and encapsulation efficiency (Y_4_). Each factor was evaluated at low (−1) and high (+1) levels and three central points (0) were added to detect curvature. ANOVA analysis and Pareto charts were used to demonstrate the influence of each factor on the response. The results are shown in Appendix A.

##### Factorial Design 2^k^

Based on the screening study results, the three main effects obtained (S/C/P concentration, drug concentration, and speed rate during injection) were carried out in a two-level factorial design with three factors. The variables that were not statistically significant or did not have a major impact on the evaluated effects were fixed at some value, according to the description depicted in Table 7.

The factorial design considered tests of power and sample size. In total, 20 experimental runs were carried out divided into 4 blocks, with 2 replicates each and 4 central points, with a base design of 3.8. The twenty runs were made and analyzed using Minitab 17 (Appendix A).

##### Central Composite Design (CCD)

The CCD was obtained from an extension of the factorial design to which axial points, called star points, points were added to obtain the response surface plot. To improve the statistical power, eight experimental runs (two central and six axial points) were added to the factorial design previously made. The CCD optimizes the response and allows a better understanding of the system and its variables. A mapping of each variable was performed and, depending on the surface response, it was possible to select the optimal configuration for each factor. The new eight runs (CCD) were analyzed through Minitab 17 (Appendix A).

#### 4.2.4. Optimized Formulation: Niosome Characterizations

After the risk analysis and experimental design, the optimized formulation was characterized by several techniques to determine the morphological and physical characteristics, particle size, drug encapsulation, drug release, and in vivo and in vitro activity.

##### Particle Size and Polydispersity Index (PDI)

To establish the particle size and determine the polydispersity index, dynamic light scattering (DLS) was performed. For particle size, PDI, and zeta potential, a Zetasizer Nano Zen 3600 with scattering angle of 173°; water viscosity of 0.8872; and refractive index of 1.496 at 25 °C and zeta dip cell were used, respectively. Three independent samples from each sample were measured at room temperature. The niosomal dispersion was not diluted.

##### Superficial Charge

The electrokinetic surface was determined by means of particle charge. It was measured with a Mutek PCD 03 particle charge detector. A sample of niosomes was diluted with deionized water (2.5 mL sample/7.5 mL water).

##### Encapsulation Efficiency (EE)

Encapsulation efficiency allows for the quantification of the encapsulated drug. Briefly, a sample was centrifuged at 12,150× *g* at 4 °C for one hour. The amount of unencapsulated CasIII-ia in the supernatant was measured spectrophotometrically (UV–visible diode array spectrophotometer 8452A) at λ= 294 nm. For each batch, a blank of niosomes was prepared with the same concentration of niosomes, but without the CasIII-ia. A calibration curve (range 8 × 10–6 M- 8 × 10–4 M) was constructed for the quantification of the percentage of encapsulation. The encapsulation efficiency was calculated according to the following equation: (1)$$\mathrm{EE}=\frac{\mathrm{amount}\;\mathrm{of}\;\mathrm{encapsulated}\;\mathrm{CasIII}-\mathrm{ia}}{\mathrm{total}\;\mathrm{amount}\;\mathrm{of}\;\mathrm{CasIII}-\mathrm{ia}\;\mathrm{used}\;\mathrm{for}\;\mathrm{the}\;\mathrm{preparation}\;\mathrm{of}\;\mathrm{niosome}}\times 100$$

##### Electronic Microscopy

Niosome morphology was analyzed by transmission electron microscopy (JOEL JEM-2010) for negative staining and by scanning transmission electron microscopy (Bruker Nano GmbH D-12498). One drop of the sample was placed on a carbon continued-film nickel grid and it was dried at room temperature. Finally, one drop of uranyl acetate 1% *w*/*v*5 was placed on the same grid.

##### Fourier Transform Infrared Spectroscopy (FTIR) and Attenuated Total Reflectance Infrared Spectroscopy (ATR-FTIR)

FTIR and ATR-FTIR were performed to verify the encapsulation of CasIII-ia. The obtained niosomes/CasIII-ia, blank niosomes, and free CasIII-ia were characterized with the classical pellet KBr method by FTIR (Nicolet Avatar 320, Thermo Scientific). Spectral transmittance over the spectral range from 4000 to 400 cm^−1^ was evaluated. 

For ATR-FTIR, a lyophilized sample (niosomes/CasIII-ia, blank niosomes, and free CasIII-ia) was evaluated in the Nicolet iS5 with an iD5 ATR accessory from Thermo Scientific. Spectral transmittance over the spectral range from 4000 to 500 cm^−1^ was evaluated.

##### Thermal Analysis

Differential calorimetry scanning (DSC) and thermogravimetry analysis (TGA) were used to characterize the thermal behavior of niosomes and niosomes with CasIII-ia. For the DSC study (DSC/700 Mettler Toledo), a sample of 2 mg of lyophilized powder was crimped in a standard aluminum pan and heated from 20 to 300 °C at a heating constant rate of 10 °C/min. For the TGA study (TGA 4000 System, PerkinElmer, 100–240 V/50–60 Hz), a sample of 10 mg was weighed and heated from 10 to 550 °C at a heating constant rate of 10 °C/min under a nitrogen atmosphere.

#### 4.2.5. Physical Stability Study

To establish the physical stability of the niosomes, the optimized formulation obtained (CasIII-ia or blank niosome) was placed in a 3 mL glass vial. Each container (*n* = three) was placed in a temperature and humidity chamber, MAYASA model CEHT 12000, at 40 °C and 75% HR. The particle size, zeta potential, and polydispersity index were determined at zero and three months. The procedure was carried out as described in the Mexican Standard NOM-073-SSA1-2015.

#### 4.2.6. In Vitro Drug Release

In vitro drug release was evaluated with the dialysis bag method. A regenerated cellulose dialysis bag of 12–14 KDa (Spectrum Labs, Asheville, NC, USA) was hydrated and washed with water for half an hour. The dialysis bag was suspended in a 40 mL PBS and shook at 37 ± 0.5 °C at 160× *g*. A sample of 0.5 mL of the prepared niosomal with CasIII-ia dispersions was placed in the donor compartment. Then, samples were taken at fixed times for 35 h; the sample taken from the receiver compartment was replenished with the same volume. A sample of blank niosomes and a free CasIII-ia sample were evaluated under the same methodology. The drug release was assayed by spectrophotometry at 294 nm and the data were adjusted to zero-order, first-order, Higuchi and Korsmeyer–Peppas model.

#### 4.2.7. In Vitro Cell Assays

In vitro cytotoxicity experiments were performed to evaluate the effect of the CasIII-ia nanoformulations on hormonal-independent triple-negative breast cancer (MDA-MB-231). The process was carried out. Briefly, cells were harvested and cultivated in 96-well plates with 2 × 104 cells each and 100 µL of the medium. The plates were incubated at 37 °C and in a 5% CO_2_ atmosphere. The next day, the medium was withdrawn and100 µL of dissolutions containing niosomes, niosomes without CasIII-ia, or free CasIII-ia niosomes (range of 1–106 µM) were added. Cell cultures were exposed for 24 h or 48 h of treatment. Finally, the medium was retired, and plates were fixed and stained with sulforhodamine B to assess cell viability according to the procedure described by Skehan [46].

#### 4.2.8. In Vivo Test

The in vivo evaluation was realized using 36 5-week-old BALB/c female mice, donated by the FES Zaragoza animal house. Following the approved protocol of FEZ-CE/22-118-08, mice were implanted with 10,000 cells (4T1, metastatic breast cancer model) contained in 100 μL of serum-free RPMI that was subcutaneously injected near the mammary pad (4th pair). After the appearance of the tumor (about 10 days after implantation), treatments were administered intraperitoneally (ip). The 30 animals were divided into 5 groups, distributed as followed: (A) control; (B) glucose solution 5% (SG); niosome/SG (1.45 mg/kg) 4 doses every 4 days; CasIII-ia (6 mg/kg) 6 doses every 4 days; cis-Pt (4 mg/kg) 4 doses every 7 days; niosome/CasIII-ia (1.45 mg/kg/6 mg/kg) 6 doses every 4 days. At the end of treatment, the animals were weighed and the tumor was measured in length and width. The percentage of weight loss and toxicity was calculated for each animal.

## 5. Conclusions

The QbD tool was used for the development of the niosome/CasIII-ia with the potential for IV administration. The QTPP was defined and used as a guide for pharmaceutical development. The Ishikawa diagram and RRMA analysis were made from the selection of materials and methods for niosome formation, and the CQAs, CPPs, CMAs, and experimental designs were defined according to ICH standards of QbD for prediction and optimization of a formulation. The first experimental design was carried out (Plackett–Burman, variable screening), where it was found that the main variables to consider were drug concentration, niosome concentration, and agitation speed during the injection. These variables were evaluated in a factorial 2^3^ design. It was found that the model presented a curvature, which is the reason why the design was adjusted to a central compound design, as it was possible to optimize the formulation of CasIII-ia with the niosomal systems. Drug concentration was the most important variable for niosome formulation. The operational conditions were 697 µM of S/C/P, 32 µM of CasIII-ia, and a 170 rpm speed rate. The obtained particle size and encapsulation efficiency were predicted according to the optimization graphs, at 150 nm and 40%, respectively; the particle size was less polydisperse and there was an improved encapsulation efficiency with respect to the chitosan nanoparticle. Thermogravimetric (TGA and DSC) and spectroscopic (FTIR) studies were carried out to confirm the incorporation of the compound inside the system. The microscopy studies showed that the niosomes are spheres with homogeneous sizes. Niosomes with and without CasIII-ia are physically stable for at least 3 months under accelerated conditions. A biphasic diffusion-governed release of CasIII-ia was observed, and this modified release coincides with the results obtained in the MDA-MB-231 cell line, where a higher activity of encapsulated CasIII-ia was observed at 48 h of study, giving a better distribution of the drugs that provide a better therapeutic effect. In the in vivo studies, also, it was observed that encapsulated CasIII-ia shows lower toxicity concerning cis-Pt and nonencapsulated CasIII-ia. A reduction in toxicity could allow a higher dose to be administered to improve effectiveness. This work opens up a window of opportunity for the formulation of coordination compounds in niosomes using quality tools that allow optimization. The optimal conditions obtained through QbD may improve the scaling-up process.

## Figures and Tables

**Figure 1 ijms-23-12756-f001:**
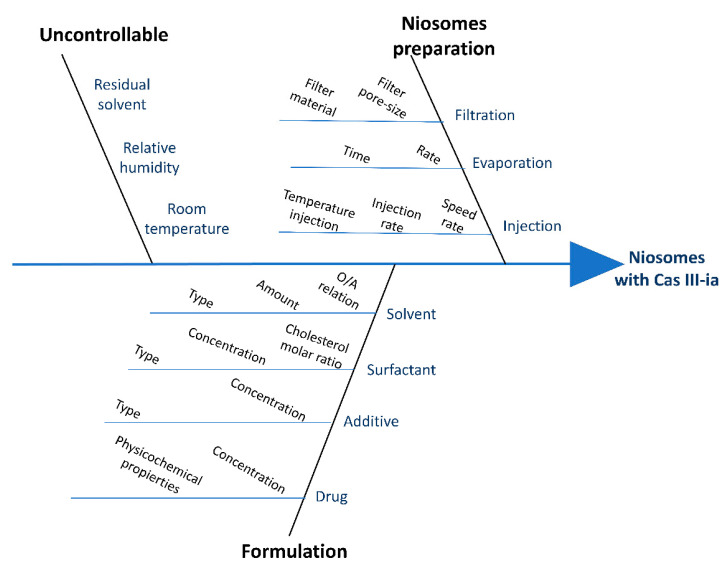
Ishikawa diagram for CasIII-ia formulation in niosomes with critical material attributes and critical process parameters.

**Figure 2 ijms-23-12756-f002:**
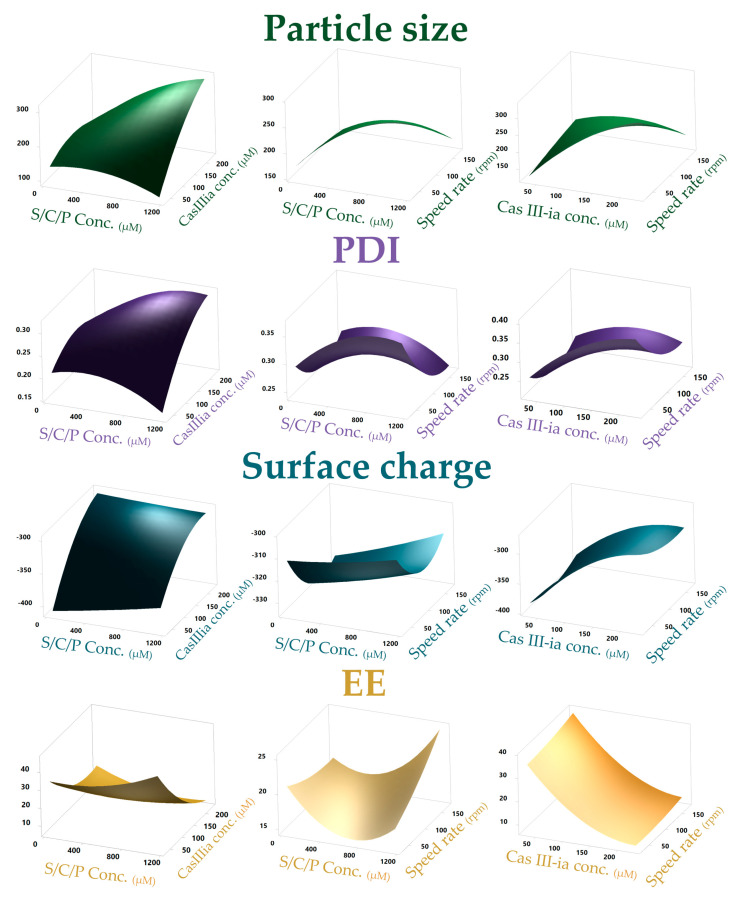
A 3D—response surface plot portraying the effect of the input variables on particle size, PDI, encapsulation efficiency, and surface charge.

**Figure 3 ijms-23-12756-f003:**
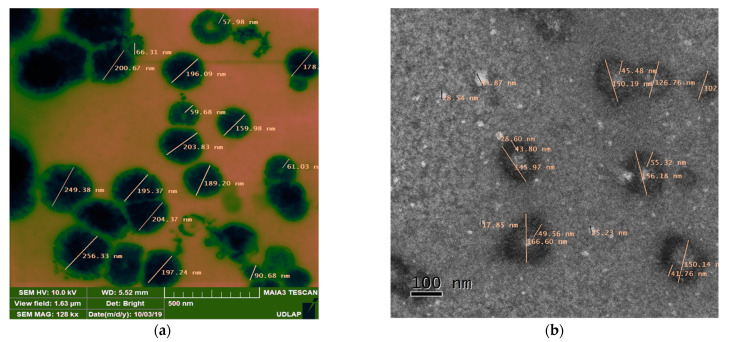
(**a**) STEM and (**b**) TEM images of the developed CasIII-ia niosomal formulation.

**Figure 4 ijms-23-12756-f004:**
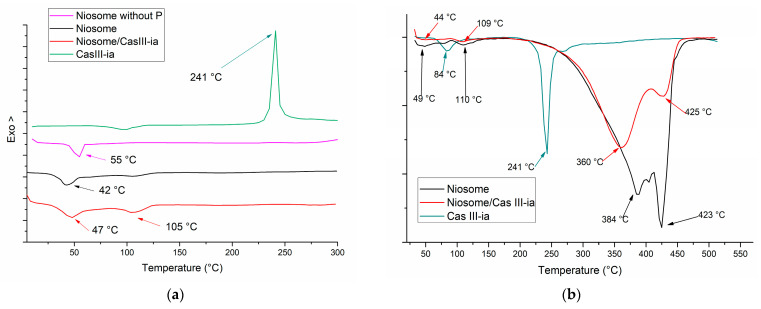
(**a**) DSC and (**b**) TGA thermogram of CasIII-ia, niosome with and without CasIII-ia.

**Figure 5 ijms-23-12756-f005:**
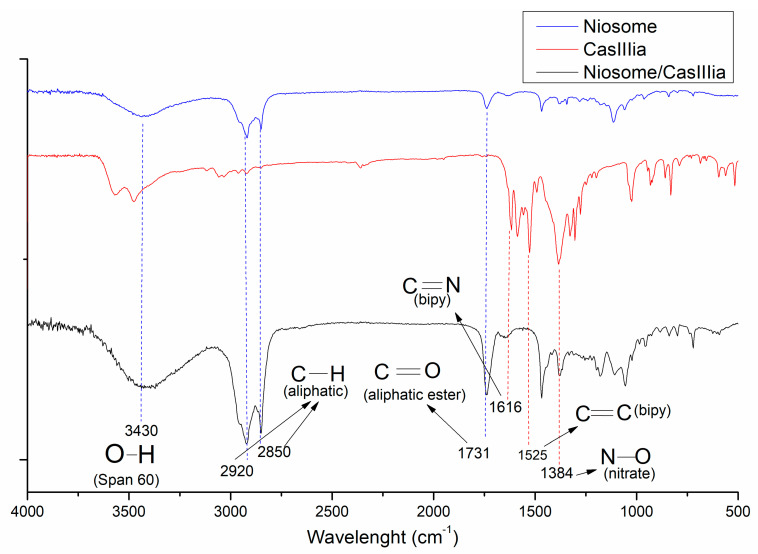
FTIR spectra of CasIII-ia (red line), niosome blank (blue line), and niosome with CasIII-ia (black line).

**Figure 6 ijms-23-12756-f006:**
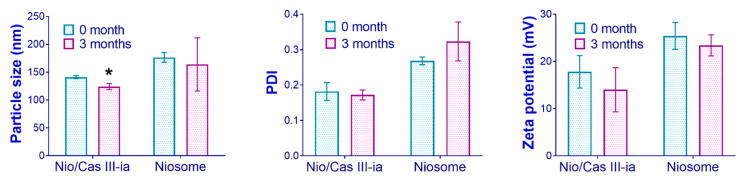
Accelerated stability studies (40 °C and 75% RH) for CasIII-ia in niosome-optimized formulation. The unpaired *t*-test was used to evaluate statistically significant groups (marked with an asterisk).

**Figure 7 ijms-23-12756-f007:**
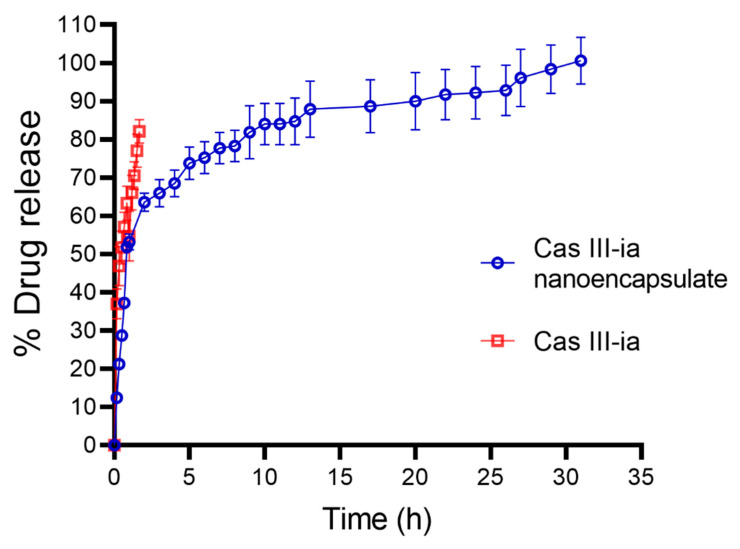
Drug release profile of nanoencapsulated CasIII-ia (blue line) and profile of free CasIII-ia (red line) (*n* = 3).

**Figure 8 ijms-23-12756-f008:**
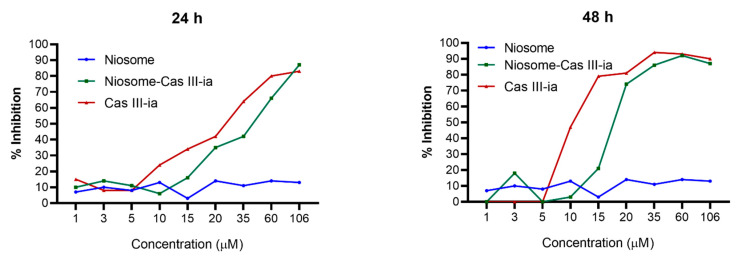
Cytotoxic effect of MDA-MB-231 cell line treated with free CasIII-ia, CasIII-ia niosomal formulation, and niosome blank at different concentrations and times.

**Figure 9 ijms-23-12756-f009:**
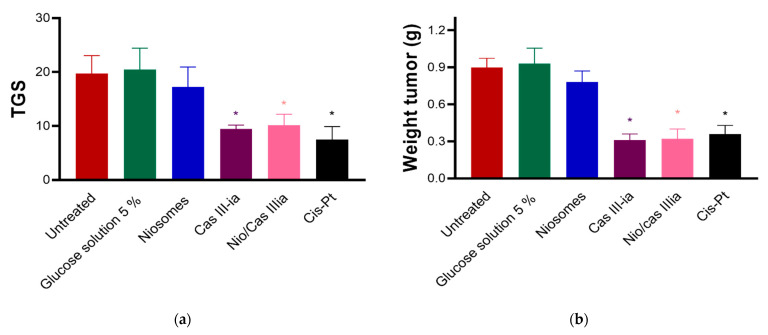
(**a**) Tumor growth speed (TGS) was calculated from tumor size based on the difference between day 21 and day 1. The unpaired *t*-test was used to evaluate statistically significant groups (marked with an asterisk). (**b**) Primary tumor weight for each treatment group. The unpaired *t*-test was used to evaluate statistically significant groups (marked with an asterisk).

**Figure 10 ijms-23-12756-f010:**
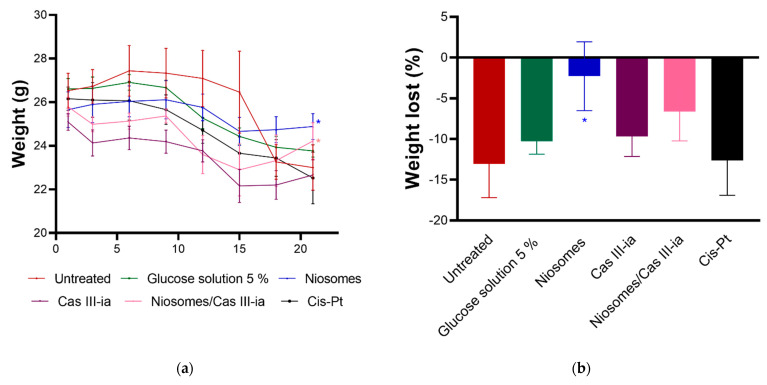
(**a**) The initial and final weight of BALB/c mice before, during, and after treatment. (**b**) Toxicity was measured by the percentage of weight loss of mice. The percent weight loss, as an indicator of toxicity, was calculated for each animal as follows: [(weight at day 21/weight at day 0) − 1] × 1000. The unpaired *t*-test was used to evaluate statistical significance among groups (marked with an asterisk).

**Table 1 ijms-23-12756-t001:** Risk analysis based on critical material attributes.

	Drug	Additive (Pluronic F127)	Cholesterol Molar Ratio	Surfactant (Span 60)	Solvent	References
CQAs						
Particle size	High	Low	Low	Medium	High	[28,29,30,31,32]
Encapsulation efficiency	High	High	Medium	High	Low	[15,16,30,33,34,35,36]
PDI	High	Low	Low	Medium	High	[31]
Zeta potential	Medium	High	Low	Medium	Low	[36,37,38]

**Table 2 ijms-23-12756-t002:** Risk analysis based on critical parameter process.

	Injection	Evaporation	Filtration	References
CQAs				
Particle size	High	Low	High	[39,40,41]
Encapsulation efficiency	Medium	Low	Medium	[39,41]
PDI	High	Low	High	
Zeta potential	Medium	Low	Medium	

**Table 3 ijms-23-12756-t003:** The niosome preparation conditions for optimized formulation (*n* = 3).

Variables Conditions	CQAs Value
697 µM of S/C/P concentration 32 µM of CasIII-ia 170 rpm of speed rate	Maximum EE (40%) Particle size of 150 nm

**Table 4 ijms-23-12756-t004:** Particle size, EE, zeta potential, and PDI niosome with and without optimized CasIII-ia (*n* = 3).

	Particle Size (nm)		EE (%)		Zeta Potential (mV)	PDI
	Observed	Predicted	Observed	Predicted		
Niosomes with CasIII-ia	150.4 ± 4.66	150 ± 19.10	39.89 ± 4.99	40.44 ± 3.87	−13.8 ± 1.25	0.234 ± 0.01
Niosome blank	215 ± 1.92	-	-	-	−23.4 ± 1.95	0.231 ± 0.01

**Table 5 ijms-23-12756-t005:** Quality target product profile for CasIII-ia nanoformulation.

QTPP	Target	Justification
Dosage form	Injectable solution	Patent same dose form. Patent solicitude (MX/a/2017/016444) [61].
Therapeutic indication	Cancer treatment (phase I clinical trial of CasIII-ia)	Casiopeinas have shown activity in tumor lines of cervical (HeLa, SiHa), breast (MCF-7), colon (HCT15), and neuroblastoma (CHP-212) cancers [9]. Moreover, CasIII-ia has shown tumor reduction in a nude mouse xenotransplanted (HCT-15) [62].
Dosage design	A modified-release dosage from a colloidal system	Modified release to favor reaching the site of action of the compound by enhanced permeation and retention effect (EPR) [63,64,65].
Route of administration	Intravenous	Same route of administration as patent. Avoid the first-pass effect [61].
Production method	Solvent ether injection; drug entrapment during niosome formation	In the solvent injection method, the use of sonication is not necessary for the reduction of particle size because it depends on experimental conditions and could be between 50 and 500 nm [28,35]. This represents a lower energy expenditure. Additionally, industrial scalability could result in better feasibility.
CQA excipients	Span 60 (sorbitan monostearate), cholesterol and Pluronic F127 (poloxamer 407)	Excipients are critical for niosome formation, entrapment efficiency, zeta potential, and blood circulation time [32,34,66].
Stability	At least 6-month shelf-life at 40 °C and 75% RH; agrees with NOM-073-SSA1-2015	Measurement of particle size and PDI is related to the quality and safety profile of the preparation [48,67].
Container closure system	Sterile glass vial	To achieve the target shelf-life and to ensure quality during storage [61,68].

**Table 6 ijms-23-12756-t006:** Critical quality attributes for CasIII-ia nanoformulation.

CQA	Target	CQA	Justification
Appearance	Colorless liquid	No	Color and appearance are not important for quality.
Particle size	150 nm	Yes	Particle sizes between 20 and 50 nm present a high efficacy and safety. Sizes below 50 nm can suffer fenestration in organs such as the liver and fast-renal clearance. On the other hand, sizes above 200 nm could present spleen retention. Finally, sizes between 100 and 150 nm remain for a longer time in blood circulation [20,48].
Zeta potential	±30 mV	Yes	The physical stability of vesicular systems depends on the total surface charge. Niosomes in solution require physical stability for storage and systemic circulation [21,48].
Polydispersity index (PDI)	Low	Yes	Niosomes with high PDI values modify the drug release profile, which may increase the toxicity of the drug.
Encapsulation efficiency	Maximum	Yes	Encapsulation efficiency diminishes free casiopeina concentration and, consequently, its interaction with blood proteins.
Residual solvents	Minimum	Yes	Residual solvents are an important toxic factor. Ether ethylic is a class 3 solvent, in agreement with ICH guideline Q3C. Class 3 solvents have low toxic potential [40]. Available data indicate they are less toxic in acute or short-time studies and negative in genotoxicity studies [40].

**Table 7 ijms-23-12756-t007:** Conditions for niosome formation in factorial design 2^3^.

Evaporation	Time	60 min	Estimated time for ethylic ether evaporation
	Speed rate	50 rpm	Low agitation is a minor energetic cost
Injection	Temperature	50 °C	Phase-transition temperature (Tc) of Span 60 is 60 °C. However, cholesterol can reduce phase-transition temperature (Tc) [32].
	Injection rate	40 mL/h	Chiraz et al. reported that the injection rate has no significant effect on the mean vesicle size [72].
	Speed rate	Variable	
	Solvent volume	2.5 mL	Smaller solvent volumes can increase particle size due to fast evaporation. Higher volumes imply higher residual solvent.
Formulation	CasIII-ia concentration	Variable	
	S/C/P concentration	Variable	
	P concentration	3%	A 2.5–5% additive concentration is recommended. Higher concentrations inhibit niosome formation [73].

## Data Availability

Not applicable.

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
