# Peer review of "Development and In Vitro and In Vivo Evaluation of an Antineoplastic Copper(II) Compound (Casiopeina III-ia) Loaded in Nonionic Vesicles Using Quality by Design"

_ijms, 2022, doi:10.3390/ijms232112756_

Round 1

Reviewer 1 Report

In the manuscript entitled “Development, in vitro and in vivo evaluation of an antineo plastic copper (II) compound (Casiopeina III-ia) loaded in non ionic vesicles, using Quality by Design” the authors have attempted to develop niosomes formulation of  CasIII-ia for IV administration through a quality by design (QbD) approach. The methodology and derived conclusions from results are coherent and well-illustrated. As there are not many reports on this kind of formulation using QbD tools, I recommend publication of manuscript after considering minor corrections.

1.     Authors are suggested to discuss the novelty or originality of present work and its overall implications.

2.     Figure 2: Please add the units for independent and response variables in response surface plots

3.      The results are not clear for thermal analysis (2.3.2). Please discuss in detail for better clarity and readability for readers.

Reviewer 2 Report

In the manuscript “Development, in vitro and in vivo evaluation of an antineoplastic copper(II) compound (Casiopeina III-ia) loaded in non-3 ionic vesicles, using Quality by Design” the authors report a novel formulation based on niosomes for Copper (II) compound CasIII-ia. The manuscript is well organized and contains interesting data, however there are some concerns and as described below:

1) In the title I did not understand the term “antineoplastic”.

2) Line 20: The sentence “As shown by this work, achieving the optimum conditions with QbD of nanoparticles could be useful in decreasing the toxicity of metal-based drugs.” is not true. The use of QbD to establish the best formulation for any kind of drugs, consequently decreasing the toxicity is evidenced in many other papers, not only in the work presented by the authors. I suggest to remove the sentence from the Abstract.

3) Line 44: A reference must be added.

4) As a general recommendation, there are too many references, and many references as papers belonging to the corresponding author.

5) Line 54 “making niosomes  a great alternative to drug delivery systems” Niosomes are drug delivery systems, thus the sentence must be rewritten.

6) The Introduction contains some paragraphs (line 62-68, line 74-84) containing definitions and common sense facts about QbD. I suggest to remove or at least shorten these parts and replace them with some relevant information about the state of the art in development of novel carriers for Casiopein compounds, other QbD studies.

7) Line 92 ‘’ For the nanoformulation of CasIII-ia in niosomes, the most important CQA determined were particle size and encapsulation efficiency’’. The authors must provide an explanation.

8) Line 115 “S/C/P concentration” An explanation must be provided for the meaning of the ratio S/C/P.

9) Line 195 “the immersion of CasIII-ia in the aqueous part” The term “immersion” is not adequate, must be changed.

10) Line 239 “Whereas, the characteristic bands of Cas III-ia are not observed in the niosome spectra with Cas III-ia, suggesting the encapsulation of the compound” It is not true. The absence of the specific signal does not means that the compound is encapsulated in the niosomes; probably the concentration is too low. The authors must rewrite the explanation of FTIR data or try to improve their experimental data. Why both KBr pellet and ATR methods were used?

11) Stability study : The authors must provide an explanation why the size of niosomes decreases. The change in the value of Zeta potential could be produced by the leaking of the encapsulated drug? Stability of the CasIII-ia will be helpful, in order to evaluate the protection efficiency.  

12) Usually ,thermal analysis is used for other purposes than proving the encapsulation. I suggest to remove this part, or to review the comments in term of thermal behavior of the formulation.

13) Line 495 “The encapsulation efficiency was obtained to determine the amount of drug inside the niosomes.” The phrase must be rewritten.

14) As a general remark, a serious revision of style must be performed, there are many confusing phrases (for example line 505 “Niosome morphology properties were analyzed by Transmission Electron Microscopy (JOEL JEM-2010) for negative staining and Scanning Transmission Electron Microscopy).”

Round 2

Reviewer 2 Report

The manuscript can be published in the revised form.